# Genome-wide identification of *ABCC* gene family and their expression analysis in pigment deposition of fiber in brown cotton (*Gossypium hirsutum*)

**Na Sun, Yong-Fei Xie, Yong Wu, Ning Guo, Da-Hui Li, Jun-Shan Gao**⊙*

School of Life Sciences, Anhui Agricultural University, Hefei, PR China

* gaojsh@ahau.edu.cn

**Data Availability Statement:** All relevant data are within the manuscript and its Supporting information files.

## Abstract

ABC (ATP-binding cassette) transporters are a class of superfamily transmembrane proteins that are commonly observed in natural organisms. The ABCC (ATP-binding cassette C subfamily) protein belongs to a subfamily of the ABC protein family and is a multidrug resistance-associated transporter that localizes to the tonoplast and plays a significant role in pathogenic microbial responses, heavy metal regulation, secondary metabolite transport, and plant growth. Recent studies have shown that the ABCC protein is also involved in the transport of anthocyanins/proanthocyanidins (PAs). To clarify the types and numbers of ABCC genes involved in PA transport in *Gossypium hirsutum*, the phylogenetic evolution, physical location, and structure of *ABCC* genes were classified by bioinformatic methods in the upland cotton genome, and the expression levels of these genes were analyzed at different developmental stages of the cotton fiber. The results showed that 42 *ABCC* genes were initially identified in the whole genome of upland cotton; they were designated *GhABCC1-42*. The gene structure and phylogenetic analysis showed that the closely related *ABCC* genes were structurally identical. The analysis of chromosomal localization demonstrated that there were no *ABCC* genes on the chromosomes of AD/At2, AD/At5, AD/At6, AD/At10, AD/At12, AD/At13, AD/Dt2, AD/Dt6, AD/Dt10, and AD/Dt13. Outside the genes, there were *ABCC* genes on other chromosomes, and gene clusters appeared on the two chromosomes AD/At11 and AD/Dt8. Phylogenetic tree analysis showed that some ABCC proteins in *G. hirsutum* were clustered with those of *Arabidopsis thaliana*, *Vitis vinifera* and *Zea mays*, which are known to function in anthocyanin/PA transport. The protein structure prediction indicated that the GhABCC protein structure is similar to the AtABCC protein in *A. thaliana*, and most of these proteins have a transmembrane domain. At the same time, a quantitative RT-PCR analysis of 42 *ABCC* genes at different developmental stages of brown cotton fiber showed that the relative expression levels of *GhABCC24*, *GhABCC27*, *GhABCC28*, *GhABCC29* and *GhABCC33* were consistent with the trend of PA accumulation, which may play a role in PA transport. These results provide a theoretical basis for further analysis of the function of the cotton *ABCC* genes and their role in the transport of PA.

**Funding:** This research was supported by the National Natural Science Foundation of China (No.31672497) and Key Research and Development Plan of Anhui province (No. 202004e11020002). The funding bodies played no role in the design of the experiment and data collection, analysis, or preparation of the manuscript.

**Competing interests:** The authors have declared that no competing interests exist.

## Introduction

Cotton (*Gossypium spp.*) is an economically important crop that is cultivated worldwide. Cotton fiber is a necessity for daily life and is an important raw material for the textile industry. Colored cotton is an eco-friendly textile raw material that does not need to be dyed, bleached or otherwise treated to obtain a certain color in the process of making textiles. The common varieties of colored cotton are brown cotton and green cotton, and brown cotton is the primary cultivated variety [1]. However, colored cotton has a number of disadvantages, such as low yield, short fiber length, low color and genetic instability of the pigment [2], which limits its application and marketing. Therefore, it is urgently important to elucidate the molecular mechanism governing colored cotton fiber pigmentation formation. At present, there are in-depth studies on the metabolic pathways of brown cotton fiber pigments. The key functional genes and regulatory factors in the proanthocyanidin (PA) biosynthesis pathway have been well studied [3], but the mechanism underlying transport and oxidative polymerization has not been elucidated to date.

Studies have found that PAs in plant cells are stored in the large central vacuole; therefore, the polymerization of PAs may occur in the vacuole [4]. Some researchers have shown that some key enzymes in the PA metabolism pathway, such as ANR, ANS, and DFR, are located in the cytoplasm [5], and hypothesized that these three enzymes may perform anthocyanin and flavan-3-ol biosynthesis in a certain area of the cytoplasm [6, 7]. Studies have shown that in *Arabidopsis thaliana* seed coat cells, epicatechin and cyanidin are initially synthesized in the cytoplasm, are transported to the vacuole by transporters, and finally aggregate to form multimers in the vacuole [8]. Previous studies have shown that during the transport of epicatechin and catechin to the vacuole, four types of proteins are involved in the transport process: ABCC protein, MATE protein (TT12), GST protein (TT19) and P-ATPase protein (TT13) [9]. A study of *A. thaliana* seed coat PAs found that TT12 is involved in the transport of epicatechin glycosides and anthocyanin glycosides, while TT19 is involved in the transport of anthocyanin glycosides and epicatechin, and the proton pump encoded by TT13 provides a concentration gradient transport epicatechin and catechin [10–12]. Studies have shown that the ABCC protein may be involved in the transport of flavonoids in some plants and can cotransport PA precursor substances with the GST protein [9].

ABC (ATP-binding cassette) is a family of ancient and large transmembrane proteins that are commonly observed in natural organisms. Most ABC transporters have activities *in vivo* and relying on the energy generated by ATP hydrolysis to achieve transmembrane transport of substrates inside and outside the cell, which include amino acids, liposomes, polysaccharides, peptides, heavy metal chelates, alkaloids and drugs [13, 14]. Multidrug resistance (MDR) is the first ABC transporter identified in eukaryotes and is involved in the process of excretion of intracellular drugs to prevent excessive accumulation of drugs in cells [15, 16]. Fifty-six and 53 ABC transporters have been identified in *Drosophila* and *Bombyx mori*, respectively, in which the Bmwh3 protein of *B. mori* and the white and brown ABC proteins of *Drosophila* have been shown to have pigment transport functions [17–21]. The corn *bronze-2 (bz2)* mutant lacks a glutathione S-transferase encoded by the *bz2* gene, resulting in the inability of anthocyanins to accumulate in vacuoles. Because glutathione S-transferase has a very important effect on the activity of MRP transporter binding substrates, it is speculated that in the maize *bz-2* mutant, the MRP type ABC transporter is likely to participate in the anthocyanin transport process [22]. Studies have found that sodium orthovanadate, an inhibitor of ABC transporter, can significantly reduce the secretion of flavonoids. It is speculated that ABC transporter may be related to the secretion of flavonoids from soybean roots [23].

Plant ABC transporters were first discovered during plant detoxification, and subsequently, a large number of ABC transporters were identified in plants. To date, the functions of ABC transporters have exceeded the scope of the detoxification mechanism. Some studies have confirmed that ABC transporters play important roles in plant pathogenic microbial responses, regulation of heavy metals, and transport of secondary metabolites [24]. With the sequencing and implementation of the plant genome, the ABC transporter has been fully identified and studied in plants. At present, the number of ABC transporters identified in plants is considerably higher than those in animals or microorganisms; for instance, there are 123 ABC transporters in *Oryza sativa*, 127 ABC transporters in *A. thaliana*, and 89 ABC transporters in leguminous plants [25–27]. A large number of ABC transporters in plants may be involved in complex metabolic activities.

The ABCC protein (ATP-binding cassette C subfamily) belongs to a subfamily of the ABC protein family and is a multidrug resistance-associated transporter. Most ABCC transporters have a transmembrane domain consisting of 3 to 5 transmembrane helices and are involved in many physiological processes, such as intracellular detoxification, transport of chlorophyll metabolites, and regulation of ion channels [28]. In addition, plant ABCC transporters play an important role in the process of storing glycosides and pigment metabolites in vacuoles. Previous studies have shown that *ABCG10* regulates the expression level of isoflavones in sputum [29]; MRP3 is transformed into the leaves by constructing an interference vector and changes the color of the leaves in maize [30]; and *VvABCC1* is involved in anthocyanins in grape skins [31].

Cotton is the most important fiber crop worldwide and exhibits a wide range of varieties, among which upland cotton (*Gossypium hirsutum*) is the most widely cultivated. With the completion of whole-genome sequencing of upland cotton [32], the genome-wide database can be used for systematic screening, identification and comparative genomics research and provides a rich resource for research on the biological functions of *ABCC* gene family members. At present, the *ABCC* genes of *A. thaliana*, *Vitis vinifera*, *Zea mays* and other species have been identified, and there are many studies on the function of the model plant *ABCC* gene family. In this study, the *ABCC* gene family of upland cotton was identified by bioinformatics. The number, sequence characteristics and evolutionary relationship of *ABCC* genes in upland cotton are analyzed at the genetic level. The function of these genes is predicted by qRT-PCR and homology comparison, and the role of the genes in the transport and accumulation of pigment in brown cotton fiber is further discussed to provide a theoretical basis for breeding pigment-stable varieties of brown cotton.

## Results

### Identification of the *ABCC* gene family in upland cotton

Using bioinformatic methods, ABCC family members were screened from upland cotton. At the same time, the basic information of all *ABCC* genes was searched using the EsPAsy online website, and the physical and chemical properties of protein length, molecular weight and isoelectric point were obtained and analyzed. The results showed that 42 *ABCC* genes were obtained in upland cotton (Table 1). The ABCC protein sequences appeared significant differences, and their amino acid lengths ranged from 435 aa (*GhABCC39*) to 1624 aa (*GhABCC24*). The molecular weights ranged from 48.52 kD (*GhABCC39*) to 182.95 kD (*GhABCC24*). The isoelectric points ranged from 5.4 (*GhABCC39*) to 8.96 (*GhABCC40*). It can be observed from the basic characteristics of these ABCC proteins that the gene family varies considerably regarding gene lengths and protein properties, indicating that the members of the gene family have different characteristics and potentially play different biological roles.

**Table 1. Characteristics of *GhABCC* genes identified in *G. hirsutum*.**

| Gene name | Gene ID | Chromosomal localization | Cds Position | Numbers of Amino acid | Molecular weight (Da) | pI | Numbers of exon |
|---|---|---|---|---|---|---|---|
| GhABCC1 | CotAD_02682 | Dt_chr9 | 41311889–41317949 | 1428 | 159320.98 | 6.92 | 11 |
| GhABCC2 | CotAD_75551 | sca | 1711–6407 | 1231 | 137429.86 | 6.38 | 11 |
| GhABCC3 | CotAD_27409 | Dt_chr3 | 39527379–39532708 | 1465 | 162797.03 | 6.43 | 11 |
| GhABCC4 | CotAD_21899 | Dt_chr11 | 59886597–59892332 | 1465 | 163846.02 | 7.67 | 12 |
| GhABCC5 | CotAD_72273 | At_chr11 | 62166046–62171020 | 1298 | 144934.19 | 7.44 | 11 |
| GhABCC6 | CotAD_21900 | Dt_chr11 | 59965930–59971598 | 1431 | 160252.41 | 7.90 | 11 |
| GhABCC7 | CotAD_48319 | Dt_chr1 | 3925636–3935180 | 1540 | 170682.03 | 7.47 | 11 |
| GhABCC8 | CotAD_62215 | At_chr1 | 6160978–6165643 | 1250 | 138003.72 | 5.94 | 10 |
| GhABCC9 | CotAD_15441 | sca | 576384–581146 | 1286 | 142314.02 | 6.20 | 10 |
| GhABCC10 | CotAD_15407 | sca | 144115–149898 | 1623 | 180455.27 | 6.44 | 10 |
| GhABCC11 | CotAD_22397 | sca | 870353–878207 | 1565 | undefined | | 12 |
| GhABCC12 | CotAD_15406 | sca | 120149–125156 | 1399 | 155348.33 | 6.36 | 9 |
| GhABCC13 | CotAD_15404 | sca | 93416–98079 | 1212 | 134306.59 | 6.16 | 11 |
| GhABCC14 | CotAD_26386 | At_chr7 | 11009721–11015682 | 1504 | 169401.33 | 8.12 | 11 |
| GhABCC15 | CotAD_16330 | Dt_chr8 | 38634127–38639663 | 1540 | 173246.96 | 7.32 | 10 |
| GhABCC16 | CotAD_22657 | Dt_chr4 | 22246611–22251942 | 1508 | 169327.55 | 6.34 | 9 |
| GhABCC17 | CotAD_71461 | Dt_chr7 | 9958770–9964794 | 1524 | 171597.89 | 8.11 | 11 |
| GhABCC18 | CotAD_21687 | Dt_chr5 | 25328345–25333684 | 1487 | 167184.25 | 8.41 | 10 |
| GhABCC19 | CotAD_39801 | sca | 131301–136337 | 1415 | 158915.88 | 8.00 | 9 |
| GhABCC20 | CotAD_57277 | At_chr3 | 15120096–15122856 | 827 | 92300.62 | 7.47 | 3 |
| GhABCC21 | CotAD_13839 | Dt_chr12 | 3046390–3054079 | 1435 | 161389.18 | 6.81 | 13 |
| GhABCC22 | CotAD_57278 | At_chr3 | 15127049–15129617 | 655 | 72410.90 | 5.82 | 8 |
| GhABCC23 | CotAD_72276 | At_chr11 | 62142290–62145829 | 858 | 94684.55 | 6.07 | 12 |
| GhABCC24 | CotAD_14237 | Dt_chr8 | 3735646–3752579 | 1624 | 182946.63 | 7.51 | 27 |
| GhABCC25 | CotAD_46251 | sca | 447046–453542 | 907 | 101297.70 | 5.69 | 9 |
| GhABCC26 | CotAD_04577 | At_chr4 | 36364268–36368203 | 1012 | 113332.26 | 6.63 | 9 |
| GhABCC27 | CotAD_23475 | At_chr8 | 40396551–40404026 | 1160 | 129215.35 | 6.07 | 15 |
| GhABCC28 | CotAD_14239 | Dt_chr8 | 3694851–3704893 | 1127 | 125785.73 | 8.55 | 16 |
| GhABCC29 | CotAD_14243 | Dt_chr8 | 3607141–3614778 | 1086 | 122059.63 | 7.24 | 14 |
| GhABCC30 | CotAD_21686 | Dt_chr5 | 25335414–25338543 | 787 | 88197.96 | 6.21 | 9 |
| GhABCC31 | CotAD_23481 | At_chr8 | 40244764–40271890 | 1313 | 147940.48 | 8.40 | 21 |
| GhABCC32 | CotAD_14236 | Dt_chr8 | 3763762–3771149 | 1086 | 121250.91 | 5.69 | 14 |
| GhABCC33 | CotAD_57900 | At_chr8 | 36145902–36153374 | 1031 | 115657.03 | 6.07 | 13 |
| GhABCC34 | CotAD_62212 | At_chr1 | 6136757–6138947 | 658 | 72785.47 | 7.55 | 3 |
| GhABCC35 | CotAD_44535 | At_chr9 | 45977082–45988622 | 999 | 110655.69 | 5.60 | 23 |
| GhABCC36 | CotAD_72272 | At_chr11 | 62224591–62226533 | 451 | 50419.82 | 8.64 | 7 |
| GhABCC37 | CotAD_01493 | Dt_chr9 | 60823161–60843906 | 1072 | 119006.68 | 6.04 | 26 |
| GhABCC38 | CotAD_43419 | sca | 4322–8619 | 606 | 67994.50 | 6.20 | 11 |
| GhABCC39 | CotAD_74385 | At_chr1 | 61527124–61529014 | 435 | 48521.00 | 5.40 | 7 |
| GhABCC40 | CotAD_49953 | At_chr11 | 61728158–61729963 | 533 | 59826.99 | 8.96 | 3 |
| GhABCC41 | CotAD_23476 | At_chr8 | 40366247–40379559 | 1078 | 121925.83 | 8.71 | 25 |
| GhABCC42 | CotAD_62211 | At_chr1 | 6115177–6125803 | 676 | 74854.98 | 5.57 | 5 |

## Phylogenetic analysis of the *ABCC* gene family

To better clarify the genetic relationship between monocotyledonous and dicotyledonous species, a phylogenetic tree was constructed according to the ABCC sequences from *Z. mays* (19),

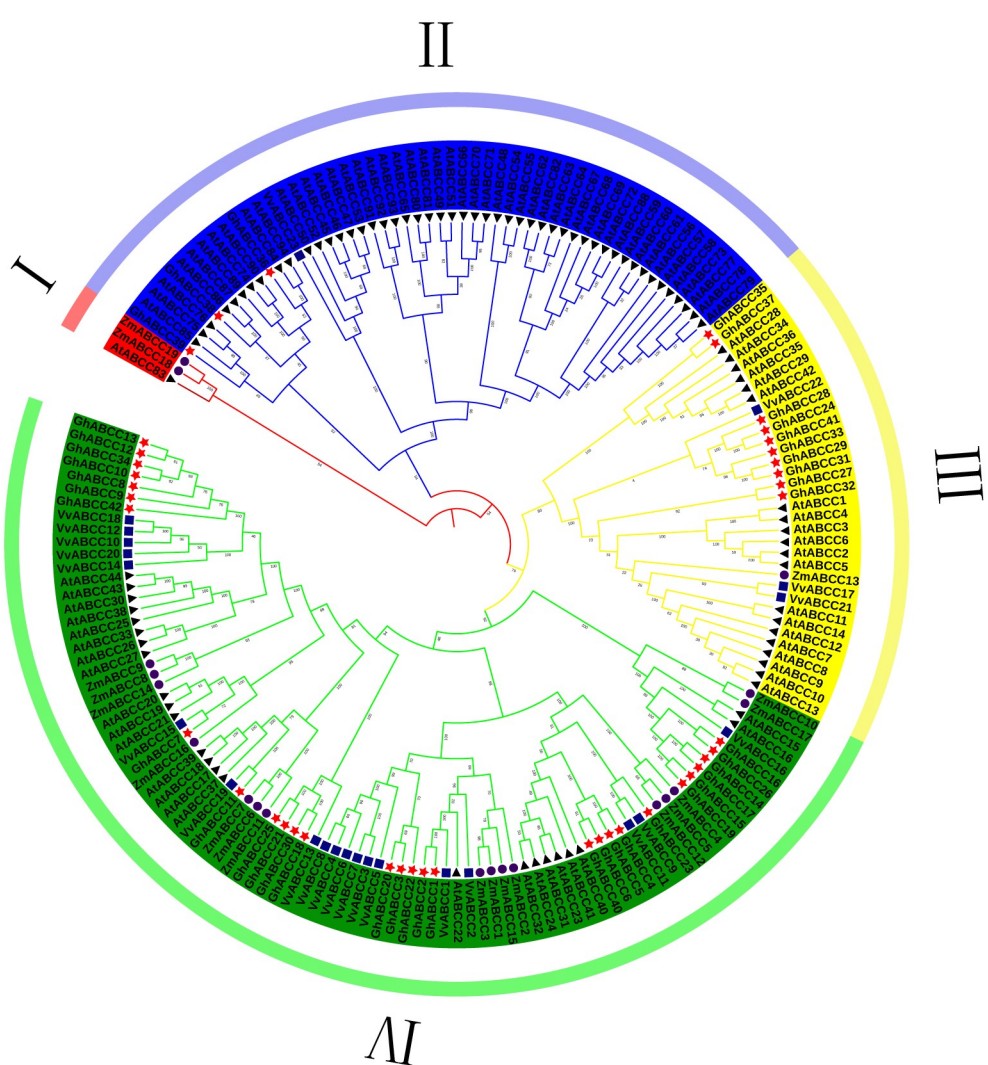

**Fig 1. Phylogenetic tree of ABCC proteins from *G. hirsutum*, *A. thaliana*, *V. vinifera* and *Z. mays*.** The tree was generated with MEGA 7.0 software (1000 bootstrap replicates) using the neighbor-joining method, Different colors indicate different subfamilies of ABCC.

*G. hirsutum* (42), *A. thaliana* (94) and *V. vinifera* (23) (Fig 1). According to the classification method described by Sun et al. [33], the *GhABCC* gene family was divided into four subfamilies: I, II, III, and IV. The results indicated that more *ABCC* genes in cotton and *A. thaliana* are clustered together, and the evolutionary relationship between them is closely related. The *AtABCC1* and *AtABCC2* genes in *A. thaliana* have been demonstrated to be involved in the transport of PAs [34, 35]; therefore, we hypothesize that members of subfamily III in *G. hirsutum* may be involved in the transport of anthocyanins in vacuoles.

## Characteristics of *GhABCC* gene structures

In this study, to better understand the evolutionary relationship of *GhABCC* genes, a phylogenetic tree was constructed with the ABCC proteins from *G. hirsutum* (Fig 2). As shown in Fig 2, there are 15 homologous pairs in the *GhABCC* gene, of which 11 pairs of bootstrap

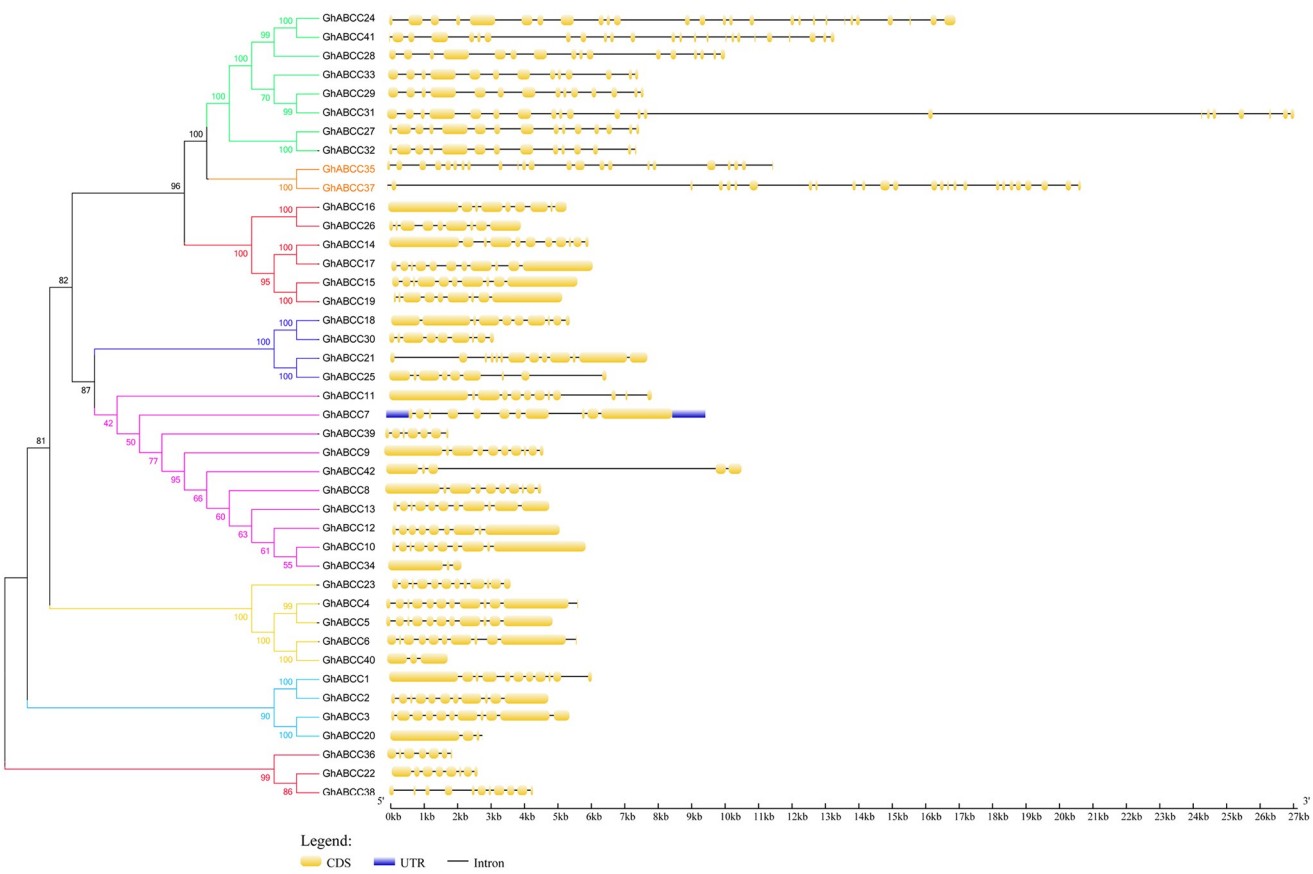

**Fig 2. Phyletic evolution and gene structure of the *ABCC* gene family in *G. hirsutum*.** The exon-intron structure map of the upland cotton ABCC family was obtained using the GSDS online tool. Cotton ABCC family phylogenetic tree (left) and gene structure (right). Yellow boxes, blue boxes and black lines represent exons, 5' or 3' untranslated region (UTR) and introns, respectively.

values are 100, indicating that the 11 pairs of *GhABCC* genes are closely related. The gene sequences of members of the same subfamily are very similar, indicating the possibility of their similarity of functions, reflecting the conservation of the *GhABCC* gene during evolution.

## Distribution of conserved motifs of *GhABCCs*

Twenty conserved sequences of ABCC protein in upland cotton were identified by the online software MEME. Also, it can be observed that all *GhABCCs* contain motifs 1, 2, 3 and 7 (Fig 3), and the conserved regions of different subfamilies are different, indicating that these motifs may have some specific functions (Fig 4). The similarity and difference of gene structure and conserved motifs reflect the relative conservation of the *GhABCC* gene family in the lengthy evolutionary process and the diversity generated for adapting to the environment.

## Chromosome localization of *GhABCC* genes

To further study the effect of gene evolution on the GhABCC gene family, chromosome localization of *ABCC* genes was analyzed using Mapinspect software in upland cotton. The results showed that *ABCC* genes were primarily distributed on At1, Dt1, At3, Dt3, At4, Dt4, Dt5, At7, Dt7, At8, Dt8, At9, Dt9, At11, Dt11 and Dt12 chromosomes in upland cotton, and most of

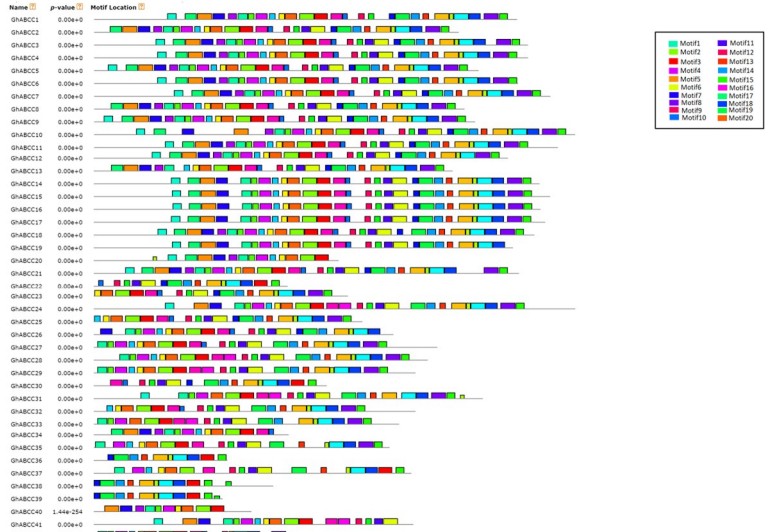

**Fig 3. Conserved motif compositions of *ABCC* genes from upland cotton.** Twenty putative conserved motifs were elucidated using MEME with complete protein sequences. All motifs have been labeled by different colors.

them were in the middle and lower parts of chromosomes; only a small number were in the upper part (Fig 5). Forty-two *ABCC* genes were randomly distributed, among which 5 *ABCC* genes were distributed on the Dt8 chromosome and were the most abundant. It is generally believed that a 200-kb nucleotide group with more than three genes is considered a gene cluster. There is a gene cluster on chromosome DT8 in upland cotton, which may encode structural genes that catalyze different steps of the same metabolic pathway. Tandem gene replication is a process in which DNA molecules replicate one or more adjacent copies, which achieve the evolution of gene families through high-frequency gene production and death. According to the definition of gene tandem replication, *GhABCC*14, 16, 26; *GhABCC*14, 15, 17; *GhABCC*4, 5; *GhABCC*6, 40; *GhABCC*24, 41; *GhABCC*27, 32; and *GhABCC*29, 31 have gene tandem replication.

## Protein secondary structure prediction and subcellular localization analysis of *GhABCCs*

Using the online analysis tool SOPMA to make predictions, it was determined that the secondary structure of ABCC protein is mainly alpha helix and random coils, and the proportion of extended strand and beta turn is relatively small (Table 2). Using Cell-PLoc 2.0 software, it was found that ABCC proteins were mainly located on the cell membrane, and only *GhABCC24, 27, 28, 29, 31, 32, 33* were located on the vacuole membrane (Table 2). It can be seen from the evolutionary tree that most of these 7 genes are in subfamily III (Fig 1); therefore, it is hypothesized that genes in subfamily III may play important roles in the transport of anthocyanins/PAs.

## Accumulation of PAs and expression analysis of *ABCC* genes in brown cotton

The different developmental stages of fiber in white cotton and brown cotton were observed. Before the boll stage, there is no difference in appearance between brown cotton and white

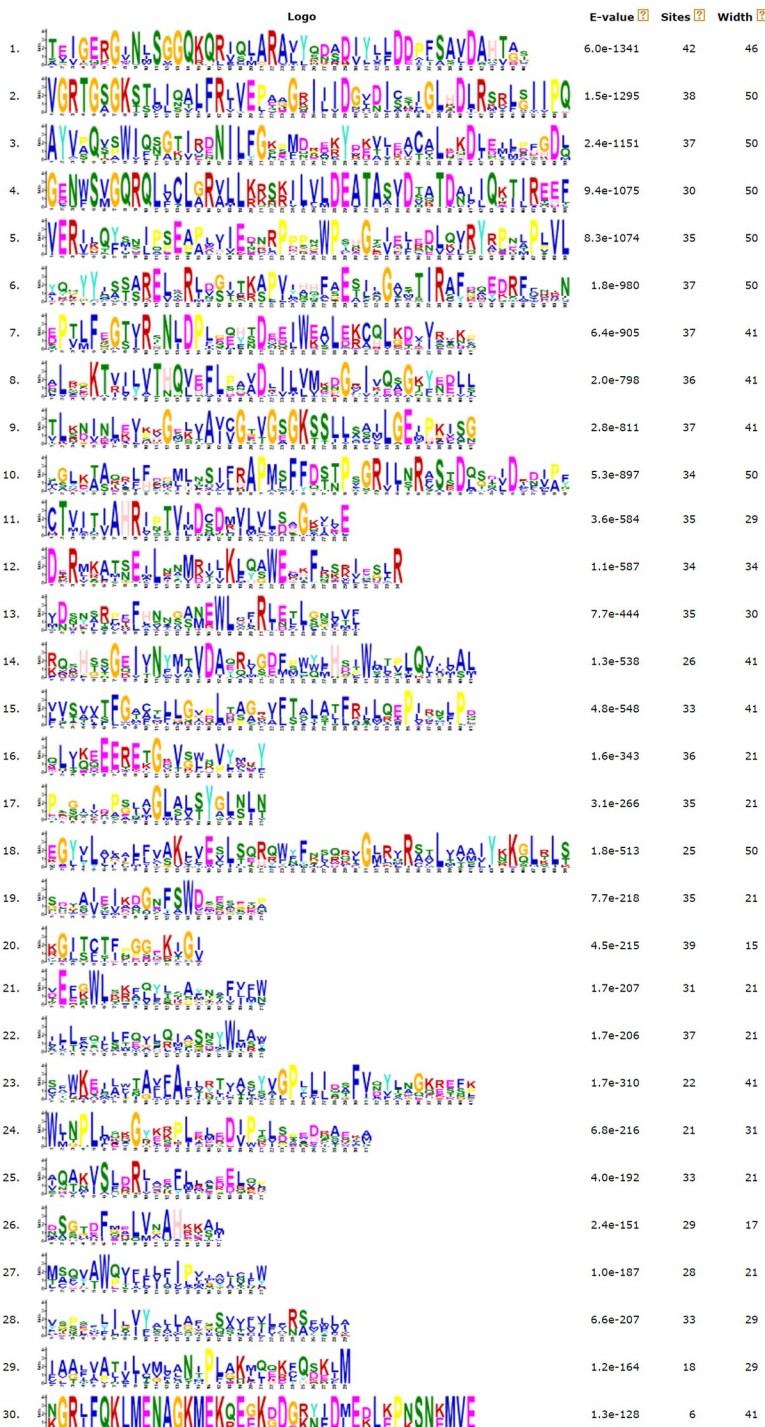

**Fig 4. Distribution of conserved motifs of ABCC proteins in *G. hirsutum*.** The x-axis indicates the conserved sequences of the domain. The height of each letter indicates the conservation of each residue across all proteins. The y-axis is a scale of the relative entropy, which reflects the conservation rate of each amino acid.

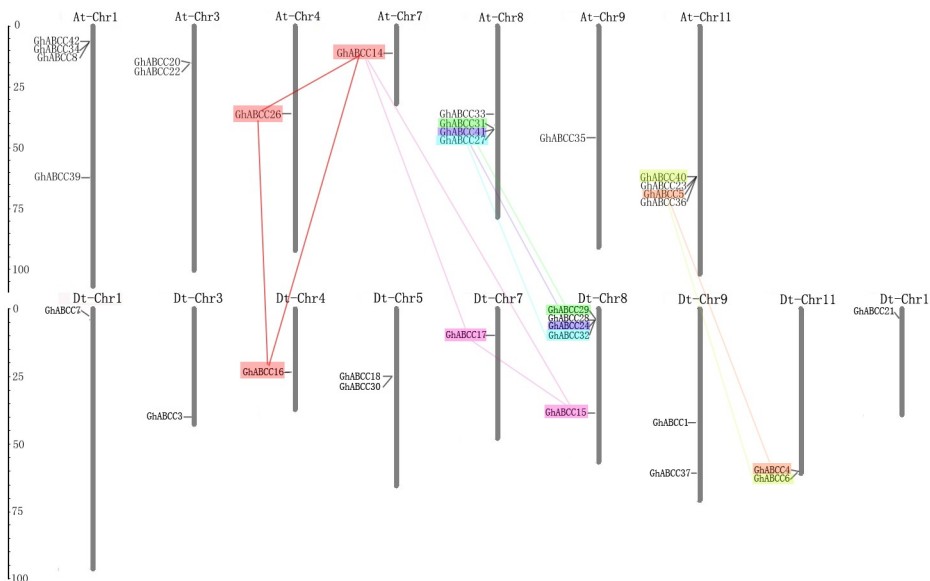

**Fig 5. Chromosomal localization of *ABCC* genes in *G. hirsutum*.** The chromosome numbers are indicated at the top of each bar, while the size of a chromosome is indicated by its relative length. The unit on the left scale is Mb, and the short line indicates the approximate position of the *GhABCC* gene on the corresponding chromosome. Segmental duplication gene pairs are connected with color lines.

cotton; during the period of boll opening, the color of brown cotton fiber gradually darkens, and white cotton is still white (Fig 6). It can be seen that environmental conditions may promote the accumulation of PAs in the brown cotton fiber and lead to the deposition of pigments, thereby darkening the color. Therefore, the contents of PAs at different development stages of brown cotton fiber were measured. The contents of PAs increased gradually with the development of fiber, reached the highest level at 12 DPA, and thereafter decreased gradually (Fig 7). The main reason for this phenomenon may be related to the expression level of related genes in the procyanidin biosynthesis pathway.

To analyze the expression levels of *GhABCC* genes related to PA accumulation during the development stages of brown cotton fiber, fluorescence quantitative primers were designed according to the *GhABCC* gene sequences (S1 Table), and quantitative RT-PCR was performed using RNA from fibers at 6 DPA, 12 DPA, 18 DPA, 24 DPA and 30 DPA. Since the content of PA in brown cotton fiber changes significantly during these five periods [36], these periods were choosed to determine the relative expression of *ABCC* genes. The results showed that the relative expression levels of most *GhABCCs* were inconsistent with the accumulation of PAs in brown cotton fibers, while the relative expression levels of *GhABCC24*, *GhABCC27*, *GhABCC28*, *GhABCC29* and *GhABCC33* were consistent with the trend of PA accumulation (Fig 8). These *GhABCC* genes belong to subfamily III, and it was also shown that the *GhABCC* genes related to PA transport are primarily located in subfamily III according to the analyses of subcellular localization. Therefore, it is hypothesized that one or several genes of *GhABCC24*, *GhABCC27*, *GhABCC28*, *GhABCC29* and *GhABCC33* may be involved in the accumulation of PAs. In addition, these genes were compared with *A. thaliana ABCC1* through DNAMAN software. Among these genes, *GhABCC27* and *AtABCC1* had the highest homology, reaching 56.19%, and the evolutionary relationship between *GhABCC27* and *AtABCC1* was the closest in the composite phylogenetic tree (Fig 1). This result suggests that *GhABCC27* may have the same function as *AtABCC1*, which is involved in the transport of PAs.

**Table 2. Protein secondary structure prediction and subcellular localization analysis.**

| Gene name | Alpha helix % | Extended strand % | Beta turn % | Random coil % | Subcellular localization |
|-----------|---------------|-------------------|-------------|---------------|--------------------------|
| GhABCC1 | 53.43 | 17.10 | 5.99 | 23.48 | Cell membrane, Cytoplasm |
| GhABCC2 | 52.23 | 15.84 | 5.85 | 26.08 | Cell membrane, Cytoplasm |
| GhABCC3 | 50.03 | 15.70 | 5.32 | 28.94 | Cell membrane, Cytoplasm |
| GhABCC4 | 53.92 | 14.61 | 4.85 | 26.62 | Cell membrane |
| GhABCC5 | 52.62 | 15.33 | 5.32 | 26.73 | Cell membrane |
| GhABCC6 | 53.67 | 15.30 | 5.03 | 26.00 | Cell membrane |
| GhABCC7 | 50.65 | 14.61 | 5.45 | 29.29 | Cell membrane |
| GhABCC8 | 50.64 | 16.88 | 6.40 | 26.08 | Cell membrane, Cytoplasm |
| GhABCC9 | 50.08 | 15.94 | 6.38 | 27.60 | Cell membrane, Cytoplasm |
| GhABCC10 | 49.60 | 17.62 | 7.15 | 25.63 | Cell membrane, Cytoplasm |
| GhABCC11 | 50.10 | 14.95 | 6.01 | 28.95 | Cell membrane |
| GhABCC12 | 50.32 | 15.80 | 6.22 | 27.66 | Cell membrane, Cytoplasm |
| GhABCC13 | 50.00 | 16.01 | 6.60 | 27.39 | Cell membrane, Cytoplasm |
| GhABCC14 | 50.73 | 15.69 | 5.39 | 28.19 | Cell membrane |
| GhABCC15 | 49.94 | 16.36 | 5.00 | 28.70 | Cell membrane |
| GhABCC16 | 50.99 | 15.25 | 5.31 | 28.45 | Cell membrane |
| GhABCC17 | 51.38 | 15.35 | 5.25 | 28.02 | Cell membrane |
| GhABCC18 | 53.67 | 15.33 | 5.51 | 25.49 | Cell membrane |
| GhABCC19 | 50.39 | 15.41 | 5.02 | 29.19 | Cell membrane, Cytoplasm |
| GhABCC20 | 54.41 | 11.37 | 3.63 | 30.59 | Cell membrane |
| GhABCC21 | 52.33 | 15.61 | 5.64 | 26.41 | Cell membrane, Cytoplasm |
| GhABCC22 | 51.60 | 16.49 | 6.11 | 25.80 | Cell membrane |
| GhABCC23 | 43.94 | 19.58 | 6.76 | 29.72 | Cell membrane |
| GhABCC24 | 57.20 | 13.36 | 4.00 | 25.43 | Vacuole |
| GhABCC25 | 45.20 | 19.85 | 7.06 | 27.89 | Cell membrane |
| GhABCC26 | 48.42 | 19.27 | 5.53 | 26.78 | Cell membrane, Cytoplasm |
| GhABCC27 | 54.74 | 15.09 | 4.74 | 25.43 | Vacuole |
| GhABCC28 | 52.71 | 16.33 | 5.68 | 25.29 | Vacuole |
| GhABCC29 | 55.06 | 15.29 | 4.97 | 24.68 | Vacuole |
| GhABCC30 | 49.30 | 19.06 | 6.35 | 25.29 | Cell membrane |
| GhABCC31 | 51.94 | 15.84 | 4.87 | 27.34 | Vacuole |
| GhABCC32 | 55.25 | 13.26 | 5.71 | 25.78 | Vacuole |
| GhABCC33 | 53.83 | 16.68 | 5.43 | 24.05 | Vacuole |
| GhABCC34 | 54.56 | 12.61 | 4.26 | 28.57 | Cell membrane, Cytoplasm |
| GhABCC35 | 51.35 | 15.42 | 4.90 | 28.33 | Cell membrane |
| GhABCC36 | 48.34 | 21.73 | 6.65 | 23.28 | Cell membrane |
| GhABCC37 | 51.96 | 16.04 | 5.32 | 26.68 | Cell membrane |
| GhABCC38 | 47.85 | 18.15 | 8.09 | 25.91 | Cell membrane |
| GhABCC39 | 52.41 | 19.31 | 6.21 | 22.07 | Cell membrane |
| GhABCC40 | 56.10 | 17.26 | 4.13 | 22.51 | Cell membrane |
| GhABCC41 | 54.17 | 14.38 | 3.53 | 27.92 | Cell membrane |
| GhABCC42 | 43.20 | 18.20 | 4.73 | 33.88 | Cell membrane, Cytoplasm |

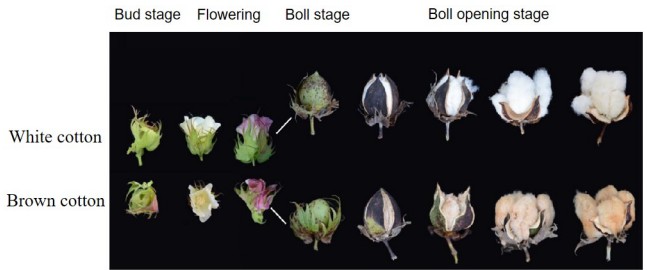

**Fig 6. The phenotypes of white fiber and brown fiber at different stages of pigment deposition.** The photographs are taken from the budding stage to the end of the boll opening stage during different growth and development periods of cotton.

## Analysis of cis-acting elements of *GhABCC27* promoter

The online promoter element prediction tool PlantCARE was used to analyze the cis-acting elements of *GhABCC27*. It was found that in addition to the basic core elements of the promoter such as CAAT-box and TATA-box, the promoter also includes a large number of elements involved in the light response (AE-box, ATCT-motif, Box 4, GT1-motif, LAMP-element, TCT-motif and chs-CMA1a), as well as plant abiotic stress inducing elements (ARE and MBS) and gibberellin response elements (GARE-motif and P-box). It has the same cis-acting elements as the GST transporter *GhTT19*, so it is speculated that *GhABCC27* may have the same transport function as *GhTT19*. In addition, it also includes a cis-acting regulatory element related to endosperm expression (GCN4_motif) and a cold-responsive cis-acting element and other cis-acting elements (Table 3). These results indicate that the expression of *GhABCC27* gene in brown cotton may be regulated by external environmental conditions such as light, plant hormones, and adversity stress.

## Discussion

In recent years, there have been many reports on the structure and function of each member of the ABC gene family, but few reports on cotton have been published. Plant ABCC

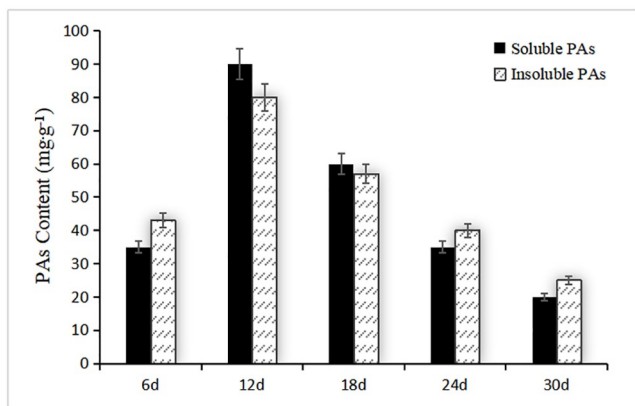

**Fig 7. PA content at different development stages of brown cotton fibers.** Abscissa indicates different days post anthesis of cotton fifibers, and ordinate indicates PA content, the error bars indicate SE.

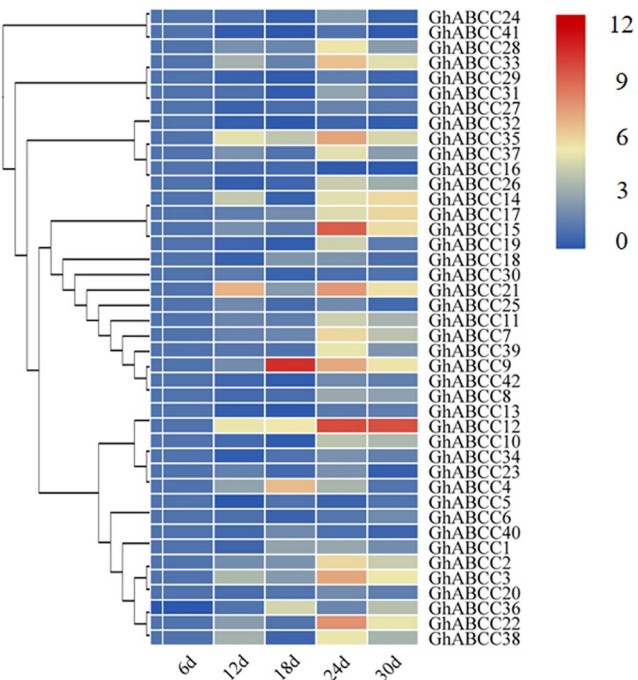

**Fig 8. Expression patterns of *ABCC* genes in *G. hirsutum*.** Relative expression levels of the *ABCC* genes at different development stages of brown cotton fiber. The relative expression level was calculated using the $2^{-\Delta\Delta Ct}$ method. Different colors represent expression level; 0 and 12 indicates the expression level, different colors represent expression level; red indicates high expression, and blue indicates low expression.

**Table 3. Functional prediction of cis acting elements of *GhABCC27* promoter.**

| Element name | Sequence (5′–3′) | Copy number | Function |
|---|---|---|---|
| AE-box | AGAAACTT | 1 | part of a module for light response |
| ARE | AAACCA | 1 | cis-acting regulatory element essential for the anaerobic induction |
| ATCT-motif | AATCTAATCC | 1 | part of a conserved DNA module involved in light responsiveness |
| Box 4 | ATTAAT | 1 | part of a conserved DNA module involved in light responsiveness |
| GARE-motif | TCTGTTG | 1 | gibberellin-responsive element |
| CAAT-box | CCAAT | 5 | common cis-acting element in promoter and enhancer regions |
| CGTCA-motif | CGTCA | 2 | cis-acting regulatory element involved in the MeJA-responsiveness |
| GCN4_motif | TGAGTCA | 1 | cis-regulatory element involved in endosperm expression |
| GT1-motif | GGTTAA | 1 | light responsive element |
| LAMP-element | CTTTATCA | 2 | part of a light responsive element |
| LTR | CCGAAA | 1 | cis-acting element involved in low-temperature responsiveness |
| MBS | CAACTG | 1 | MYB binding site involved in drought-inducibility |
| P-box | CCTTTTG | 1 | gibberellin-responsive element |
| TATA-box | TATA | 7 | core promoter element around -30 of transcription start |
| TCT-motif | TCTTAC | 1 | part of a light responsive element |
| TGACG-motif | TGACG | 2 | cis-acting regulatory element involved in the MeJA-responsiveness |
| chs-CMA1a | TTACTTAA | 2 | part of a light responsive element |

transporter plays an important role in the process of vacuolar storage of glycosides and pigment metabolites, and is generally named MRP in related research reports [37]. Studies have shown that both *ZmMrp3* and *ZmMrp4* of corn are involved in the accumulation of pigments in vacuoles [30]. Among the 129 ABC genes, 15 ABC genes encoding MRP proteins have been identified [38], but only the structure and function of the MRP gene family has been studied in *A. thaliana*. The initial discovery of the plant MRP gene was due to the observation that the entry of glutathione compounds into the vacuole was dependent on ATP for energy, rather than the proton potential difference inside and outside the membrane [39]. Both *AtMRP1* and *AtMRP2* have glutathione-conjugated transport activity, and *AtMRP2* is more active than *AtMRP1* in Arabidopsis [40]. There is evidence that *AtABCC1* functions as an anthocyanin transporter that depends on GSH without the formation of an anthocyanin-GSH conjugate [41].

In this study, the *ABCC* gene families of Arabidopsis, grape, maize and upland cotton were identified and analyzed at the genomic level. 42 *ABCC* genes were identified in upland cotton. The high number of family members determines the diversity and specificity of ABCC gene family functions. According to the homology and domains of the conserved sequences of ABCC transporters, *GhABCCs* are divided into 4 subfamilies, and the members of each subfamily are named systematically. The increase in the number of genes among species is considered to be the way to promote the evolution of species. The main way to increase the number of gene families is gene replication. Gene replication can be divided into intragene replication and intergene replication. The upland cotton genome undergoes several gene duplication events in the process of replication, and the copies of these genes are usually free from selection pressure [42]. This phenomenon not only guarantees the evolution of upland cotton but also enriches the diversity of the upland cotton gene family. Previous studies have shown that epicatechin is synthesized in the cytoplasm, transported by transporters to the vacuole, aggregates and accumulates in the vacuole. Previous studies have found that the *AtABCC1* and *AtABCC2* genes play an important role in the accumulation of PAs in Arabidopsis seed coats [34, 35]. *VvABCC17* has also been shown to be located in the vacuolar membrane and participate in the transport of glycosylated anthocyanins. From the evolutionary tree, we can clearly observe that *AtABCC1*, *AtABCC2* and *VvABCC17* [41] and *G. hirsutum* are clustered in subfamily III. The subcellular location predicts that the genes of subfamily III are primarily located on the vacuole membrane (Table 2). Therefore, we hypothesize that members of subfamily III may also play an important role in the synthesis of brown cotton fiber PAs. Through fluorescence quantitative RT-PCR analysis, it was found that the relative expression levels of *GhABCC24*, *GhABCC27*, *GhABCC28*, *GhABCC29* and *GhABCC33* were consistent with the trend of PA accumulation in brown cotton fibers (Figs 7 and 8). Therefore, we hypothesize that these genes may be related to the transport of PAs. Sequence alignment analysis of these genes with Arabidopsis *ABCC1* through DNAMAN software shows that *GhABCC27* and *AtABCC1* have the highest homology, and the evolutionary relationship was the closest in the phylogenetic tree (Fig 1). This result suggests that *GhABCC27* may have the same function as *AtABCC1*, which is involved in the transport of PAs.

The online promoter element prediction tool PlantCARE was used to analyze the cis-acting elements of *GhABCC27*. The results showed that in addition to the basic core elements of the promoter such as CAAT-box and TATA-box, the promoter also includes a large number of elements involved in the light response (AE-box, ATCT-motif, Box 4, GT1-motif, LAMP-element, TCT-motif and chs-CMA1a), as well as plant abiotic stress inducing elements (ARE and MBS) and gibberellin response elements (GARE-motif and P-box) (Table 3). These results indicate that the expression of *GhABCC27* gene in brown cotton may be regulated by external environmental conditions such as light, plant hormones, and adversity stress. In addition,

because *GhABCC27* and GST transporter *GhTT19* have the same cis-acting elements, *GhABCC27* may have the same function of transporting PAs as *GhTT19*.

## Materials and methods

### Experimental material and genome databases

Zongcaixuan 1, which has natural brown fiber, is a kind of upland cotton line bred by our laboratory. This line is planted in the high-tech agricultural park of Anhui Agricultural University (Hefei, PR China) in accordance with normal field management. The genome-wide database for upland cotton is available from the website (http://mascotton.njau.edu.cn) [32]. The *A. thaliana* genome data are from the database (http://www.arabidopsis.org/).

### Identification of *ABCC* gene in upland cotton genome

A local database of the whole genome sequence of *G. hirsutum*, *V. vinifera*, *A. thaliana* and *Z. mays* was established using DNATOOLS software. Using the amino acid sequence of *A. thaliana AtABCC1* (PF00005.27) as a query, TblastN (E-value = 0.001) sequence alignment was performed on the established local database of amino acid sequences of four species [42, 43], and the ABCC family genes were initially screened. The results were tested in the Pfam [44] database (http://pfam.xfam.org/) and CDD [45] (https://www.ncbi.nlm.nih.gov/Structure/cdd/wrpsb.cgi) to screen the ABCC sequence of the gene signature domain (ABC_membrane, ABC_tran). Use ExPASy [46] (http://www.expasy.org/) to analyze the amino acid sequence online to determine the isoelectric point (PI) of amino acid, the molecular weight (MW) of the protein, and the instability coefficient.

### Construction of the phylogenetic tree of the *GhABCC* gene family

The ClustalW tool involved in the MEGA7.0 [47] software was utilized to perform multiple sequence alignment according to the amino acid sequence of the upland cotton *ABCC* gene, and the phylogenetic tree was subsequently constructed by the neighboring method (NJ, Neighbor-Joining). Branch support values indicate nonparametric bootstrap values (in percentages of 1000 replicates). Meanwhile, a total of 178 ABCC protein sequences from *A. thaliana*, *V. vinifera* and *Z. mays* were obtained from the NCBI database (https://www.ncbi.nlm.nih.gov/). A comprehensive analysis was performed, and a composite phylogenetic tree of the ABCC proteins was drawn by the above method.

### Analyses of chromosomal localization, gene structure and conserved motifs

The position information of each *ABCC* gene on the chromosome was obtained from the cotton genome database, and the physical position of these genes on the chromosome was mapped using MapInspect (http://mapinspect.software.informer.com) software. The exon-intron structure map of the upland cotton ABCC family was obtained using the GSDS [48] online tool (http://gss.cbi.pku.edu.cn/). According to the obtained protein sequence, the MEME [49] online analysis tool (http://meme.sdsc.edu/) was employed to analyze the motif pattern of the upland cotton ABCC family protein.

### Signal peptide detection and subcellular localization prediction

The secondary structures of the ABCC proteins were predicted using the online analysis tool SOPMA (https://npsa-prabi.ibcp.fr/cgi-bin/npsa_automat.pl?page=/NPSA/npsa_sopma.html). The Cell-PLoc 2.0 online analysis platform can perform subcellular localization of proteins of eukaryotes, humans, plants, gram-positive bacteria, gram-negative bacteria and

viruses. The subcellular localizations of these proteins were predicted by the Cell-PLoc 2.0 [50] algorithm (http://www.csbio.sjtu.edu.cn/bioinf/Cell-PLoc-2/).

## Gene expression analysis of GhABCC

The primers for fluorescent quantitative RT-PCR were designed based on the selected ABCC subfamily gene sequences in upland cotton (S1 Table). The RNA of the fibers of different developmental stages of brown cotton was extracted, and the RNA was reverse transcribed into cDNA and subjected to qRT-PCR analysis. The qRT-PCR volume was 20 μL, including 10 μL of SYBR premix Ex Taq enzyme, 2 μL of cDNA, and 0.8 μL of upstream and downstream primers. The reaction procedure was as follows: 50˚C for 2 min; 40 cycles of 95˚C for 30 s, 95˚C for 5 s, and 60˚C for 20 s followed by 72˚C for 10 min. The UBQ7 gene was used as an internal reference [51]. Each sample was subjected to three biological replicates, and the relative expression levels were calculated using the $2^{-\Delta\Delta Ct}$ method [52].

## Determination of PA content

According to methods from Ikegami [53], the soluble and insoluble PAs of brown cotton fifibers at different developmental stages (6 DPA, 12 DPA, 18 DPA, 24 DPA, and 30 DPA) were extracted, and the content of PAs was determined by spectrophotometry according to the standard curve of catechins, which were used as controls [54]. For each experiment, three biological replicates were executed.

## Analysis of promoter cis-acting elements

The sequence of about 2000 bp before the ATG upstream of the *GhABCC27* gene is obtained from the upland cotton genome, which is the promoter sequence. Use PlantCARE [55] (http://bioinformatics.psb.ugent.be/webtools/plantcare/html/) software to conduct bioinformatics analysis of *GhABCC27* promoter and predict its main cis-acting elements.

## Conclusions

In this study, the ABCC gene family was analyzed by bioinformatics in different species, and 23, 19, 94 and 42 *ABCC* genes were identified in *V. vinifera*, *Z. mays*, *A. thaliana* and *G. hirsutum*, respectively. These genes were analyzed to determine their phylogenetic evolution, chromosome localization, gene structure, orthologous genes and gene expression patterns. ABCC genes could be divided into 4 major subfamilies in upland cotton; members of the same subfamily had the same or similar gene structure, and there was a large gap in the gene structure of members of the different subfamily. In the phylogenetic tree, through homology comparison with *A. thaliana* and prediction of subcellular location, it was preliminarily determined that the genes related to PA transport were located in subfamily III. By comparing the expression of *ABCC* subfamily genes and PA content at different developmental stages in brown cotton fiber, five candidate genes related to PA transport were screened out. Through bioinformatic analyses, the role of ABCC family genes in upland cotton in the process of pigment transport was initially explored, which may help to establish a theoretical foundation for further research studying the function of *ABCC* genes and analyzing the molecular mechanism of PA transport across membranes in brown cotton.

## Supporting information

**S1 Table. Sequences of primers used for RT-PCR in this study.**
(DOCX)

## Acknowledgments

We thank the Master student Anane. G. Owusu for providing assistance with English writing. We thank Prof. Yan Meng (College of Life Sciences, Anhui Agricultural University) for critical comments to the manuscript.

## Author Contributions

**Conceptualization:** Na Sun, Jun-Shan Gao.

**Data curation:** Na Sun, Ning Guo.

**Formal analysis:** Yong-Fei Xie, Yong Wu.

**Funding acquisition:** Jun-Shan Gao.

**Methodology:** Na Sun, Yong Wu.

**Project administration:** Da-Hui Li, Jun-Shan Gao.

**Software:** Na Sun, Yong-Fei Xie.

**Validation:** Na Sun, Yong-Fei Xie, Yong Wu.

**Writing – original draft:** Na Sun.

**Writing – review & editing:** Da-Hui Li, Jun-Shan Gao.

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
