## [Decision Letter · Decision Letter 0]

15 Dec 2020

PONE-D-20-32250

Genome-wide Identification and Expression Analysis of ABCC Gene Family in Gossypium hirsutum

PLOS ONE

Dear Dr. Gao,

 Your manuscript entitled "Genome-wide Identification and Expression Analysis of ABCC Gene Family in *Gossypium hirsutum* " has been evaluated by the reviewers. After careful consideration, we feel that your manuscript has merit but requires major changes as suggested by the reviewers. Therefore, we invite you to submit a revised version of the manuscript that addresses the points raised during the review process as placed at the bottom of this letter. If you are willing to address the quries raised by reviewers, i am pleased to consider your manuscript.

reviewers comments are appended below for more details.

We look forward to receiving your revised manuscript.

Kind regards,

Pradeep Sharma

Academic Editor

PLOS ONE

2. Please amend either the title on the online submission form (via Edit Submission) or the title in the manuscript so that they are identical.

Reviewers' comments:

Reviewer's Responses to Questions

**Comments to the Author**

1. Is the manuscript technically sound, and do the data support the conclusions?

Reviewer #1: Partly

Reviewer #2: Partly

2. Has the statistical analysis been performed appropriately and rigorously? 

Reviewer #1: N/A

Reviewer #2: N/A

3. Have the authors made all data underlying the findings in their manuscript fully available?

Reviewer #1: Yes

Reviewer #2: Yes

4. Is the manuscript presented in an intelligible fashion and written in standard English?

Reviewer #1: Yes

Reviewer #2: No

5. Review Comments to the Author

Reviewer #1: The manuscript PONE-D-20-32250 entitled “Genome-wide Identification and Expression Analysis of ABCC Gene Family in Gossypium hirsutum” firstly reported the ABCC family genes in upland cotton, and performed the comprehensive and systemic bioinformatic analysis, as well as their expressions during pigment deposition stages of brown fiber development. The logic of this manuscript is clear and the content looks some interesting, while it’s truely hard to accept it as current form as some critical issues in this manuscript can not be ignored. Thus, I prefer to propose the suggestion of “Major revision” for the manuscript. All the issues as following should be seriously concerned and addressed.

1.Recommendation to modify the title as “Genome-wide Identification of ABCC Gene Family and Their Expression Analysis in Pigment Deposition of Fiber in Brown-fiber cotton”.

2.As we know that brown-fiber cotton is tetraploid cultivar (AADD), thus, it is better to number the ABCC genes like 1A, 1D, 2A, 2D, etc, for a more clear understanding of these genes. While, it is also fine to name the ABCC genes as current format GhABCC1-42.

3.All the legends of the figures are not sufficient to explain the figures.

4.Current form of Figure 6 is not suitable for publication in academic journal, a visualized heatmap might be better for reading.

5.A strongly suggested experiment for the supplement of the manuscript should be performed: a simple qRT-PCR of these ABCC genes during different stages of pigment deposition is not enough to elucidate their possible functions involved in proanthocyanidin accumulation, therefor, two parts of the experiment are advised to be added, 1) the phenotypes of brown-fibers in different stages of pigment deposition, 2) the content detection of proanthocyanidin in different brown-fiber developmental stages.

6.The cis-element analysis of promoters of the GhABCC genes could be supplemented, to better understand the possible regulatory mechanism.

7.The Discussion section should be totally modified because of the narrow descriptions. Noticing that the discussion should be rewrote tightly focusing the obtained results in the manuscript.

8.Line 202, the reason to select the materials of 6 DPA, 12 DPA, 18 DPA, 24 DPA, and 30 DPA fibers should be explained.

9.Table 3 could be listed as a supplementary file.

10.Line 233, the description of the sentence “A total of 178 ABCC genes were identified” is not accurate, actually, the manuscript identified 42 ABCC genes from G. hirsutum.

11.Other minor revisions

1)For the text descriptions, there exists amounts of short sentences in the manuscript, such as Lines 50-54, and of same words and phrases in different sentences such as “at present”, etc. Thus, the text descriptions should be carefully improved throughout the manuscript.

2)Lines 62-64， the sentence “Therefor, ……” could be deleted.

3)Line 68, of the sentence “and they......”, the word “they” could be deleted.

4)Lines 88-89, more reports of ABCC genes in plants should be cited.

5)Line 110, the words “in China” could be replaced by “worldwide”.

6)Line 111, the word “cultivated” could be modified as “planted cultivar”.

7)Line 112, the reference [30] could be replaced by a newly published article of “Huang G, Wu Z, Percy RG, Bai M, Li Y, Frelichowski JE, Hu J, Wang K, Yu JZ, Zhu Y. Genome sequence of Gossypium herbaceum and genome updates of Gossypium arboreum and Gossypium hirsutum provide insights into cotton A-genome evolution. Nat Genet. 2020 May;52(5):516-524” for citation.

8)Line 128, the words “were very different” could be modified as “appeared significant differences”.

9)Line 133, the words “play different biological roles” could be modified as “potentially play different biological roles”.

10)Line 140, a symbol between the word IV and The lost.

11)Line 152, the sentence “indicating that their functions may be similar” could be modified as “indicating the possibility of their similarity of functions”.

Reviewer #2: 

The authors provided genome-wide analysis of GhABCC gene family in upland cotton is a good work, while the description should be improved and some comments also should be modified as follows:

1. The quality of figures needs improvement;

2. Some of the statements are awkward, such as “The resulting amino acid sequence…” in Line 273,

3. I suggest put the table 4 as supporting information;

4. Please removing AJE Completed files from the supporting information;

5. The reference for upland cotton genome you used is not incorrect.

6. In line 44, “one or several of these genes were selected for cloning and to verify their biological functions,..” This description is inaccurate, please modify it.

6. PLOS authors have the option to publish the peer review history of their article (what does this mean?). If published, this will include your full peer review and any attached files.

---

## [Author Response · Author response to Decision Letter 0]

20 Jan 2021

Dear Reviewers,

Thank you and the reviewers for your comments concerning our manuscript entitled “Genome-wide Identification and Expression Analysis of ABCC Gene Family in Gossypium hirsutum” (PONE-D-20-32250). These comments are all valuable and very helpful for revising and improving our manuscript. We have revised the manuscript carefully according to the reviewers’ comments, and have made some corrections marked in the revised manuscript. We hope that these modifications will be satisfactory for you and the reviewers.

We appreciate you very much for your efforts on our manuscript. It would also be of our great pleasure if our work is acceptable for publication in PLOS ONE.

We are looking forward to hearing from you soon.

Sincerely yours,

Jun-Shan Gao

---

## [Editor Report · Decision Letter 1]

25 Jan 2021

Genome-wide Identification of ABCC Gene Family and Their Expression Analysis in Pigment Deposition of Fiber in Brown Cotton (Gossypium hirsutum)

PONE-D-20-32250R1

Dear Dr. Gao,

We’re pleased to inform you that your manuscript has been judged scientifically suitable for publication and will be formally accepted for publication once it meets all outstanding technical requirements.

Kind regards,

Pradeep Sharma

Academic Editor

PLOS ONE

---

## [Editor Report · Acceptance letter]

21 Apr 2021

PONE-D-20-32250R1 

Genome-wide Identification of ABCC Gene Family and Their Expression Analysis in Pigment Deposition of Fiber in Brown Cotton *(Gossypium hirsutum)*

Dear Dr. Gao:

I'm pleased to inform you that your manuscript has been deemed suitable for publication in PLOS ONE. Congratulations! Your manuscript is now with our production department. 

Kind regards, 

on behalf of

Dr. Pradeep Sharma 

Academic Editor

PLOS ONE